# Analysis of Changes in the Amount of Phytosterols after the Bleaching Process of Hemp Oils

**DOI:** 10.3390/molecules27217196

**Published:** 2022-10-24

**Authors:** Andrzej Kwaśnica, Mirosława Teleszko, Damian Marcinkowski, Dominik Kmiecik, Anna Grygier, Wojciech Golimowski

**Affiliations:** 1Department of Food Chemistry and Biocatalysis, Wroclaw University of Life Sciences, C.K. Norwida Street 25, 50-375 Wrocław, Poland; 2Department of Food Technology and Nutrition, Faculty of Production Engineering, Wroclaw University of Economics and Business, Komandorska 118/120, 53-345 Wrocław, Poland; 3Department of Agroengineering and Quality Analysis, Faculty of Production Engineering, Wroclaw University of Economics and Business, Komandorska 180/120, 53-345 Wrocław, Poland; 4Department of Food Technology of Plant Origin, Faculty of Food Science and Nutrition, Poznan University of Life Sciences, Wojska Polskiego 31, 60-624 Poznan, Poland

**Keywords:** *Cannabis sativa* L., phytosterols, hemp oil, Finola, Earlina8FC, Secuieni Jubileu, bleaching earth

## Abstract

Unrefined vegetable oils from niche oilseeds are now sought after by consumers because of their unique nutritional properties and taste qualities. The color and flavor intensity of niche oils is a big problem, and their refining is not industrially feasible due to the small production scale. The study undertaken aimed analyze the effect of changing the amount of phytosterols (PSs) after the bleaching process of hemp oils of the ‘Finola’, ‘Earlina 8FC’ and ‘Secuieni Jubileu’ varieties. Cold-pressed (C) and hot-pressed (H) crude vegetable oils were bleached with selected bleaching earth (BE) at two concentrations. The post-process BE was extracted with methanol. The amount of PSs in the crude oils and the extract after washing the BE with methanol was analyzed by GC (gas chromatography). The study shows that the bleaching process did not significantly affect the depletion of PSs in the oils. Trace amounts of PSs remain on the BE and, due to methanol extraction, can be extracted from the oil. The conclusion of the performed research is that the bleaching of hemp oil does not cause depletion of the oil, and it significantly improves organoleptic properties. The oil bleaching process results in an oil loss of less than 2% by weight of the bleached oil, while the loss depends on the type of BE used. The study shows that the loss of phytosterols after the bleaching process averages 2.69 ± 0.69%, and depends on the type of BE used and the oil extracted from different varieties of hemp seeds.

## 1. Introduction

Vegetable oils are an essential component of human diet. On an industrial scale, rapeseed oil, sunflower oil and olive oil are produced in Europe [1]. Vegetable oil production is increasing due to its versatile uses, including biofuel production [2]. In addition to the large-scale production of refined vegetable oil, niche oils are being produced, with demand increasing due to their unique properties [3]. Niche vegetable oils are obtained from oilseeds, vegetable seeds and fruits which have a marginal amount of raw material in the oil market. The high oily content of the seeds prompts producers to press them. For example, almond oil is extracted from almond tree seeds with an oil content of 35–40% [4], argan oil is extracted from argan seeds with an oil content of about 50–55% [5], black seed oil from nigella seeds contains 28–36% oil [6], Ricinus oil from Ricinus seeds is extracted with an oil content of 40–50% and oil from flaxseed with an oil content of 30–40% [7], oil from grapeseed is extracted with an oil content of 7–20 % [8], oil from thistle seeds with an oil content of 20–25% [9], oil from plum seed with an oil content of 23–30% [10], oil from tomato seed with an oil content of 33–38% [11], and oil from hemp seed with an oil content of 28–35% [12].

Commercially produced vegetable oils undergo refining processes to remove desirable and undesirable components. The purification of crude vegetable oil from undesirable substances that negatively affect the quality and shelf life of the fat (phospholipids, free fatty acids (FFA), phytosterols (PS), waxes, oxidation products, water, aromatic compounds, pigments) is carried out through a refining process that consists of degumming, deacidification, bleaching and deodorization [13,14,15]. As a result of the degumming process, phospholipids are mainly removed [16]. The efficiency of this process is high, with up to 98% reduction in phospholipids contained in the oil [17]. The presence of FFA in the oil results in the degradation of fats. Their removal is performed by neutralization with alkaline solutions. The deodorization process is performed to remove light ether fractions neutralizing the characteristic odor of vegetable oil. All of the processes mentioned above require the use of a highly costly plant, the cost of which is impossible to amortize at a small scale of production. The bleaching process does not require using highly complex apparatus [18]. The process involves absorbing undesirable oil components due to BE [19]. As demonstrated by B. Laska et al. [18], the type of BE and the amount significantly affect the efficiency of phospholipid reduction. The efficiency is comparable to degumming and is a 99% reduction in phospholipids from oil [18]. The effectiveness of this process in removing other contaminants such as chlorophyll or carotenoids and solids has also been confirmed [20,21]. The effectiveness of BE depends on the mineral used, e.g., bentonites [22] or kaolinites [23], and the method of activation [24]. Membrane methods are an alternative for reducing oil phosphorus [25].

Bleaching earth (BE) is a natural mineral enriched with elements of various properties. Commonly used minerals are bentonites consisting mainly of montmorillonite, e.g., attapulgite clay, diatomaceous earths’ natural mineral rock deposit. Enriched minerals with various elements, such as silicon, magnesium, or activated carbon increase their effectiveness [26,27,28]. The BE is widely used, for example, for wine clarification, water filtration, vegetable oil, and biodiesel [29]. Bentonite has been observed to affect the elemental composition of wine without changing its food qualities [30]. It also affects the structure of vegetable oils, depending on the parameters of the bleaching process [31].

Hemp oils, unrefined, are a valuable source of vitamins, mineral salts, PSs and carotenoids [32]. The ratio of n−3 to n−6 acids is 3:1, and beneficial for human health [33]. The PSs contained in the oil have a positive effect on human health [34]. The content of hemp oils depends on many factors, including both the variety of the plant and the process of obtaining the oil [12]. The regular consumption of PSs can result in a significant decrease in cardiovascular disease [34].

In this study, three varieties, ‘Finola’, ‘Earlina 8 FC’ and ‘S. Jubileu’, were selected from among 600 varieties of hempseed [35]. Based on a literature analysis of little-known varieties of hempseed, the products formed from the ‘Fedora’ variety have also been well described [36]. It was hypothesized that with the help of bleaching earth, it would be possible to isolate PS. Through the bleaching process, the organoleptic parameters of the oil are improved, and due to the high cost of the process, the possibility of isolating PSs from the BE after bleaching was analyzed. The results presented here are part of a large project in which the effects of the bleaching process on fatty acid profile, elemental composition, and the degree of color change and reduction in carotenoids and chlorophyll composition were analyzed.

## 2. Results and Discussion

### 2.1. Results of Statistical Descriptive Analysis of Mass Balance of Hemp Oil Bleaching Process

Raw vegetable oils, depending on the raw material, are characterized by their individual intense odor and taste. The refining process of industrially produced oils nullifies these characteristics. The bleaching positive performed in the study is one of the refining technology processes performed to change the color (Figure 1). The presented research results are part of the research aimed at verifying the significance of the changes occurring in oils due to the bleaching process. The post-positive effect is a change in organoleptic properties with the preservation of valuable vitamins and components of vegetable oil.

Niche oils, which include hemp oil [3], are marketed in unrefined form. They are characterized by intense taste, color and texture properties. The use of BE, which is commonly used to clarify wine and juices [30], has been used to improve oil properties. The use of BE removes pigments in the form of chlorophyll, carotenoids and phospholipids from the oil [18]. Valuable components of vegetable oils are PS, which is desirable in oils; therefore, the purpose of the study was to verify whether the use of the bleaching method will reduce their share in the hemp oil and whether it is possible to extract them from BE. An important parameter for the producer is the loss of oil due to the separation of BE. A comparative analysis of oil weight loss after bleaching was performed and correlated with the amount of remaining extract after extraction from the land with methanol (Figure 2).

The type of BE used significantly affects oil losses after the bleaching process. From the analysis of the results, it can be concluded that the highest oil losses, at a level in proportion to the amount of earth used of about 1% and 2%, use P1 and P3 earth. P2 earth, attapulgite clay pH-modified, had the lowest absorption and a high efficiency of absorption of compounds from oil. The almost 20-fold difference in the mass of compounds absorbed by the BE and extract was due to the use of methanol for extraction. This difference is due to the amount of fat that remained in the BE after extraction with methanol. In our experience, the complete opposite effect was with the use of hexane, which leached out oil while the carotenoids and chlorophyll were permanently bound to the BE.

### 2.2. Results of Statistical Analysis of Process Factors on Phytosterol Profile in Hemp Oils after Bleaching

After the bleaching process, the collected results of the analysis of the individual PS contents in the BE extract were given a statistical analysis by multivariate variance MANOVA. It turns out that the independent variables in the form of the type of oil, the temperature of the seeds subjected to pressing, and the type of earth in all cases had a significant effect on the individual PS contents in the extracts. Analogous to evaluating PSs in the oil, Table 1 analyzes the same composition. The results of the statistical analyses are shown in Table 1.

In order to illustrate the individual PS contents extracted from the oil, the total amount of PSs in the weight of the extract is shown in Figure 3. Statistical analysis was performed using Duncan’s method, which shows that all variables had a significant effect on the individual PS contents extracted from the oil.

The BE was characterized by varying PS adsorption strength (Figure 3). From the experiment, it is clear that regardless of the variety of hemp from which the oil was extracted, the method of pressing and the dosage of BE used, the extracts obtained after washing P2 BE with methanol contained the least PS. According to the specifications shown in Table 1, this BE is a pH-modified attapulgite clay, with the lowest value of this parameter among all BE used in the experiment. The acid activation of BE is considered a key factor in determining the efficiency of the adsorption of compounds, including PSs and their changing molecular structure, as suggested by the analysis of phytosterol content the in extracts obtained after the purification of BE P1 and P3. They belong to the same group of adsorbents as P2 (attapulgite clay; AC), with a slightly alkaline reaction (pH = 8). The extracts obtained after P2 earth bleaching contained a maximum of 5.59 mg PS/g dm (‘S. Jubileu’/cold pressing/2.5% BE), 8.23 mg PS/g dm (‘Finola’/hot pressing/5% BE) and 13.84 mg PS/g dm (‘Earlina 8 FC’/hot pressing/2.5% BE). Compared with the PS content in samples leached from physically activated attapulgite clay (P1) and unmodified AC (P3), these values are 4–6 times lower (Table A1, Table A2 and Table A3). The use of acid-activated BE in the refining of edible oils results in the formation of steradians (SDs) from PSs via a catalyzed dehydration reaction [37,38], particularly, 3,5-stigmastadiene (stigdien; a β-sitosterol derivative), 3,5-campestadiene (camdien) and 3,5,22-stigmastatrien (stigtrien) [15,39,40]. Verleyen et al. [41] noted that the formation of these structures is related to bleaching temperature (four-fold increase in steradian concentration in the temperature range of 90–110 °C), increasing the concentration of BE and the degree of its acid activation (linear increase in SD content). In our study, the steradian content was not determined. However, it is reasonable to assume that the low concentration of PSs in P2 extracts may have resulted not so much from their poor adsorption as from advanced dehydration and the formation of PS–steroidal hydrocarbon derivatives. Moreover, as mentioned earlier (Figure 3), the use of this type of BE resulted in the lowest weight losses of hemp oils during the bleaching process, which from an economic point of view is an important advantage for industrial use. Acid-activated BE is preferred in the oil refining process because it has a higher adsorption capacity for substances considered ballast, and is more chemically active than neutral BE [42].

After the bleaching process of hemp oils, most PSs were adsorbed by physically activated attapulgite clays P1 and P3 and kerolite chemically modified diatomaceous earth (hydrated magnesium silicate; P7; Figure 3). Weaker adsorption properties were exhibited by bentonite BE, including acid-activated earth P6 (pH = 6.5). Due to its very good sorption properties, attapulgite is competitive with other clays used for the clarification of edible oils. Thermal treatment improves the cleansing capacity of AC and develops the acidity of its surface. As a result, the removal of contaminants from the oil is carried out under mild conditions with high efficiency. Kerolithic clay, on the other hand, even untreated by chemical or physical pretreatment, is considered a good adsorbent for separation purposes in many biotechnological processes [43]. Łaska-Zieja et al. [18] demonstrated the high efficiency of a 2% (m/m) addition of kerolith in reducing the phosphorus content of rapeseed oil subjected to low-temperature bleaching. Bentonite, on the other hand, is widely used in the production of bleaching earths due to its availability. In its acid-activated form, it is used by many palm oil refineries. However, the use of this form of bentonite causes the corrosion of process equipment [44] and catalyzes many chemical reactions (including isomerization, degradation and dehydration) that result in undesirable compounds in bleached oils [45].

The efficiency of edible oil refining refers to the ability of the process methods used, including bleaching, to remove undesirable substances. However, losses of vitamins, tocopherols and PS also occur at all stages of refining [46]. As our experiments showed, attapulgite clays that are not acid-activated adsorb PSs well, and their content in the extracts tested was about 25 mg/g (Figure 3). However, this was an average value for all hemp oil samples tested. Meanwhile, during the bleaching of hot-pressed ‘Earlina’ oil, the 5% addition of P3 BE resulted in the adsorption of 52.69 mg PS/g of extract (Table A2). After pressing and bleaching under identical conditions, the seed oil of the S. Jubileu variety (Table A1) had an extract of 22.13 mg PS/g, while that of the Finola variety had 21.70 mg PS/g (Table A3). In cold-pressed oils, the values were 29.69, 23.19 and 15.83 mg PS/g, respectively. This observation is significant because our previous studies show that in crude hemp oils extracted from the seeds of the aforementioned varieties of *Cannabis sativa*, the content of plant PSs was not statistically different and was about 2 mg/g of oil [35]. This suggests that the adsorption capacity of the BE tested against PS depends not only on the properties of BE and the BE dosage used (Table A1, Table A2 and Table A3; Figure 2) but also on the presence of PS companion substances in oils tested, which may affect their loss during the bleaching process. This is because the competitive adsorption between PSs and dyes contained in oils is possible, as mentioned earlier [47].

In general, increasing the dose of BE in the bleaching process of hemp oils was accompanied by an increase in PSs in the extracts (Figure 3; *p* < 0.05), although from a practical point of view, these differences were not significant. On the other hand, when considering the effect of this factor on PS adsorption by BE at the varietal level, it can be observed that in the case of BEs such as attapulgite P3 and bentonite (P5, P6) with a higher dose of BE (5%), the proportion of PS in the extracts increased in all analyzed samples, regardless of the pressing temperature of the oil subjected to bleaching (C, H). In other cases, the effect of BE addition was inconclusive. This observation confirms that the choice of BE dosage should be determined on a case-by-case basis, taking into account the physical and chemical peculiarities of the oil (especially those obtained from niche raw materials), its production technology, and the expected results of the process. This issue is important not only from the point of view of costs and accounting in business, but also from the point of view of the need to manage the used BE [48].

### 2.3. Analysis of Significant Factors of Phytosterol Profiles of Hemp Oils with the PCA Method

Principal component analysis (PCA) was applied to observe possible clusters of phytosterol content in hemp oil of the Finola, Earlina and S.Jubileu varieties bleached in different conditions. The result of the distribution of the samples depending on the differentiating factor (different variety, temperature of pressing, type of bleaching earth, and earth dose) is shown in Figure 4. The first two principal factors accounted for 88.3% (Dim1 = 72.8% and Dim2 = 15.5%) of the total variation. Factor 1 was mainly correlated with the total phytosterols content (r = 0.991), β-sitosterol (r = 0.985) and campesterol content (r = 0.978).

PCA analysis showed that most of the samples are distributed longitudinally along the Y-axis in the center of the plot. A greater scattering of samples is observed on the left–right side of the *X*-axis. When analyzing the influence of the cannabis variety (Figure 4A), the smallest dispersion is characteristic of the S. Jubileu variety. ‘Finola’ and ‘Earlina 8 FC’ oil samples are more dispersed. This is especially true of the ‘Earlina 8 FC’ variety. In addition to centrally grouped samples, we can observe two groups of outliers. The first includes five samples (rightmost of the X-axis) including samples prepared with bleaching earth P3. In the second group, there is one sample with an extreme outlier. It is a sample obtained by hot-pressing that was bleached with P2 bleaching earth in an amount of 2.5%.

When analyzing the effect of the type of bleaching earth used for the treatment process (Figure 4C), greater variation in the samples was observed when bleaching earth P2 was used. These samples were arranged vertically above the Y-axis and on the left side of the X-axis. The use of other bleaching earths did not affect the obtained effects. Most of the samples were centrally located in the plot. The exception was bleaching earth P3, where a greater horizontal dispersion was observed. During the analysis of the influence of the temperature of the oil production and the amount of bleaching earth used (Figure 4B,D), a similar scattering effect was observed. The samples were characterized by lower variability and a central distribution of the graphs and the Y-axis.

The PCA results show differences between individual oil sample’s bleaching under different conditions. However, the factors influencing the most differentiation of the samples were the hemp variety and the type of bleaching earth used, in particular, type P2.

## 3. Materials and Methods

### 3.1. Description of the Source of Hemp Oils and the Parameters of Bleaching Earth

Hot-pressed and cold-pressed oils from three varieties of hemp seed, ‘Finola’, ‘Earlina 8 FC’ and ‘S. Jubileu’, harvested in August 2021 in Poland [35], were used in the study. The seeds were divided into samples of 25 kg each after a preliminary analysis of their parameters. The press was preheated, and 5 kg of seeds were pressed to stabilize the pressing variations. Subsequently, 10 kg of seeds were measured, and the proper measurement of yield and process efficiency was made. The oil was pressed at a variable seed temperature of 20 °C (C), and the seed was heated to 60 °C (H). A sample of 2 dm^3^ was taken from each oil portion and filtered into Buchner funnels. The oil after filtration was used to analyze the bleaching process. The pressing process is described in detail in the publication by Golimowski et al. [35]. The oil was subjected to bleaching with seven types of BE, the characteristics of which are shown in Table 2. BE is a commercial product dedicated to food products.

### 3.2. GC-MS Method Description and Phytosterol Profile of Hemp Oils

Many methods are known for determining phytosterols in edible oils. In addition to the well-known GC-MS and HPLC methods, spectroscopic methods have been developed [49]. Due to our extensive experience in GC-MS, this method was chosen for the presented study. For the analysis of PSs, a total of 0.05 g of each hemp oil was used. To the samples, we added 50 µg of internal standard (5α-cholestane-Supelco, Bellefonte, PA, USA). The samples were saponified with 1 M KOH in methanol, and the unsaponifiables were extracted using a mixture of hexane and methyl tert-butyl ether (1:1, *v*/*v*). The solvent was evaporated under a nitrogen stream, and dry residues were dissolved in anhydrous pyridine (Supelco, Bellefonte, PA, USA) and silylated with BSTFA + 1% TMCS (Supelco, Bellefonte, PA, USA). The PSs were analyzed using a Hewlett-Packard 6890 gas chromatograph (Agilent Technologies. Palo Alto. CA, USA) in splitless mode with an FID detector and a DB-35MS capillary column (25 m × 0.20 mm, 0.33 μm; Agilent J&W, Folsom, CA, USA). The detector and injector were set at a temperature of 300 °C. The oven temperature was initially 100 °C for 5 min, increasing at 25°C/min to 250 °C, and then at 3 °C/min to 290 °C. The final temperature was held for 20 min. The carrier gas was hydrogen, and the flow rate was 1.5 mL/min. PS were identified by comparing their retention times with those of the standards. The PSs were determined in duplicate [50].

The subject of this study was the analysis of the effect of the bleaching process of hemp oil on the change of the phytosterol profile. Table 3 shows the phytosterol profile of the oils used for the study

### 3.3. Mass Balance of the Process and Method of Bleaching Hemp Oil

The oil obtained by seed pressing at 20 °C and 60 °C was subjected to bleaching using seven different types of BE at 2.5% and 5% m/m by weight of the oil. The BE was added to 100 g of oil at 60 °C and stirred for 10 min. Solid fractions were separated from the mixture using a Buchner funnel with a cellulose membrane and vacuum system. The BE samples were stored at −20 °C. The BE samples were then extracted with methanol in appropriate amounts of 50 g for samples where 2.5 g of BE was used and 70 g for samples where 5 g of BE was used. The uneven ratio of methanol was due to previous experiments in which it was found that these amounts of methanol were entirely sufficient to extract PS from BE. The methanol, along with the BE sample, was stirred and heated evenly to about 50 °C and then poured into a Buchner funnel with a cellulose membrane. The methanol was removed from the extract using a vacuum evaporator. Using Algorithm 1, the relative amount of absorbed compounds from the hemp oil was calculated.
(1)$$E_{s}=\frac{M_{ee}-M_{zb}}{M_{o}}\;\left[\%\right]$$
where:*E_s_*—the relative proportion of the amount of absorbed compounds from the oil (%);*M_ee_*—weight of bleaching earth after bleaching process (g);*M_zb_*— weight of bleaching earth before bleaching process (g);*M_o_*—weight of hemp oil (g).

The relative proportions of the amount of absorbed compounds from the oil are shown in Table 4.

The result of the methanol extraction was a small amount of condensed substance with a dark green color. Table 5 shows the mass of extract extracted from 100 g of hemp oil.

The resulting sample solution of methanol and extracted substances was stored in a freezer at −20 °C. A vacuum evaporator was used, the methanol was evaporated entirely from the samples, and the net extracts were analyzed for PSs. The sterol content of the BE extracts was determined using an analogous GC method. Table 5 shows the average values of the results obtained in relation to the varying use of BE.

## 4. Conclusions

The presented research results are part of a project that comprehensively analyzed the effect of different bleaching earths on the change in physicochemical parameters of hemp oils of the ‘Finola’, ‘Earlina 8FC’ and ‘S. Jubileu’ varieties. Characteristics of the seeds and oil used in the presented research were described by Golimowski et al. [35]. Niche oils, due to the small scale of their production and their large amounts of valuable nutritional compounds, are not refined. The process of blanching vegetable oil does not require the use of complex industrial apparatus and can be applied to small quantities of niche oils produced. The study shows that the use of bleaching earths up to 5% results in significant color changes without a significant reduction in the valuable phytosterols found in hemp seed oils. The bleaching process generates some loss of oil, which remains in the bleaching earth. The research shows that this amount is not small, up to 2% of the oil weight, and depends on the earth used. This is important information for niche oil producers.

The purpose of the study was to analyze the effect of the bleaching process on the change in the composition of sterols in oils. Statistical analysis and PCA analysis show that the type of bleaching earth and the variety of hemp seed have a significant effect on the change in the amount of reduced phytosterols in oil. The temperature of the pressed hemp seeds and the amount of earth used had no significant effect on the level of phytosterol reduction in the oil. In conclusion, the bleaching process does not cause significant changes in the amount of phytosterols in hemp seed oils, regardless of the variety and the conditions for obtaining oil from the seeds. The total relative value of phytosterol reduction in all cases was 2.69 ± 0.69%.

## Figures and Tables

**Figure 1 molecules-27-07196-f001:**
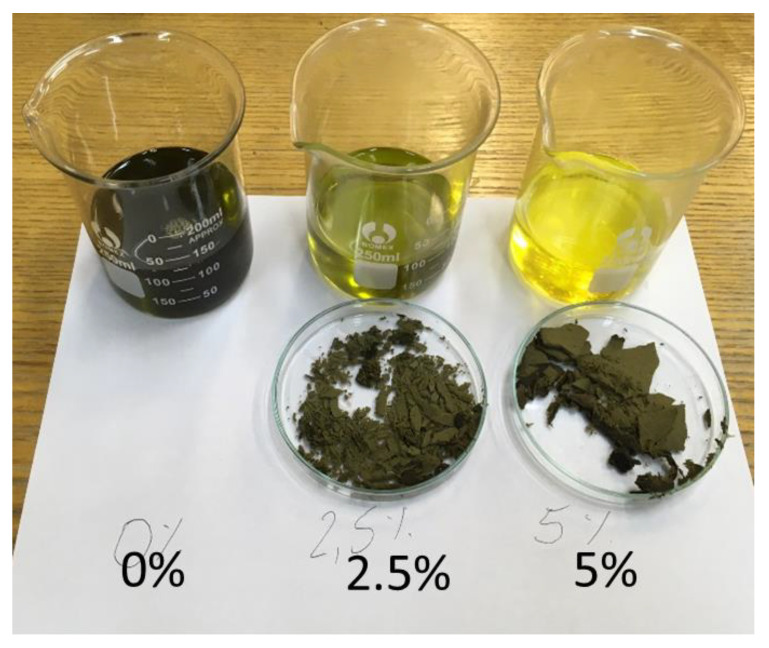
Change in the color of hemp oil as a result of the bleaching process using 2.5% and 5.0% bleaching earth m/m to oil.

**Figure 2 molecules-27-07196-f002:**
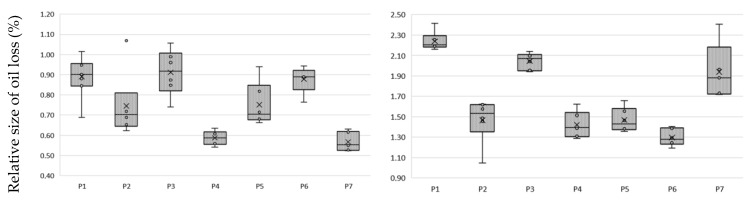
Loss of oil due to bleaching and the amount of extract extracted using earth at: (**a**) 2.5% m/m; (**b**) 5% m/m; mean and standard deviation of *n* = 6.

**Figure 3 molecules-27-07196-f003:**
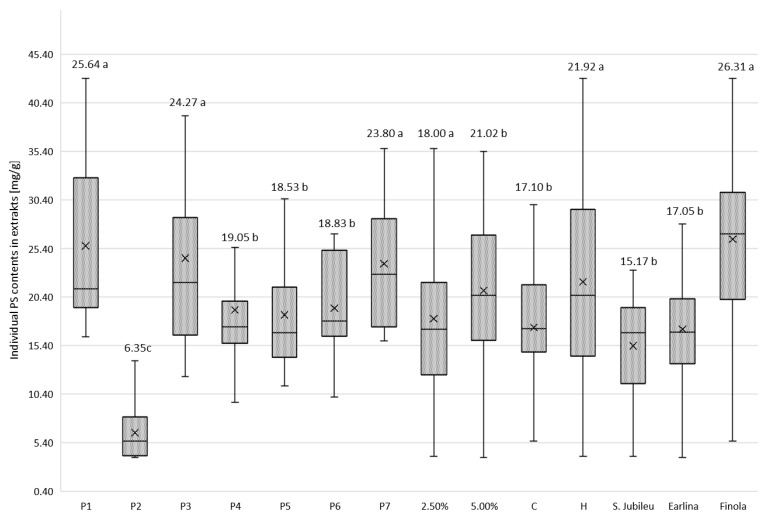
Mean values of phytosterol content (mg/g of dm) in post-bleaching methanolic extracts according to hemp variety, temperature of oil pressing, dose of bleaching earth and type of bleaching earth. a,b,c—homogeneous groups in Duncan’s test (*p* < 0.05) determined separately for each independent variable.

**Figure 4 molecules-27-07196-f004:**
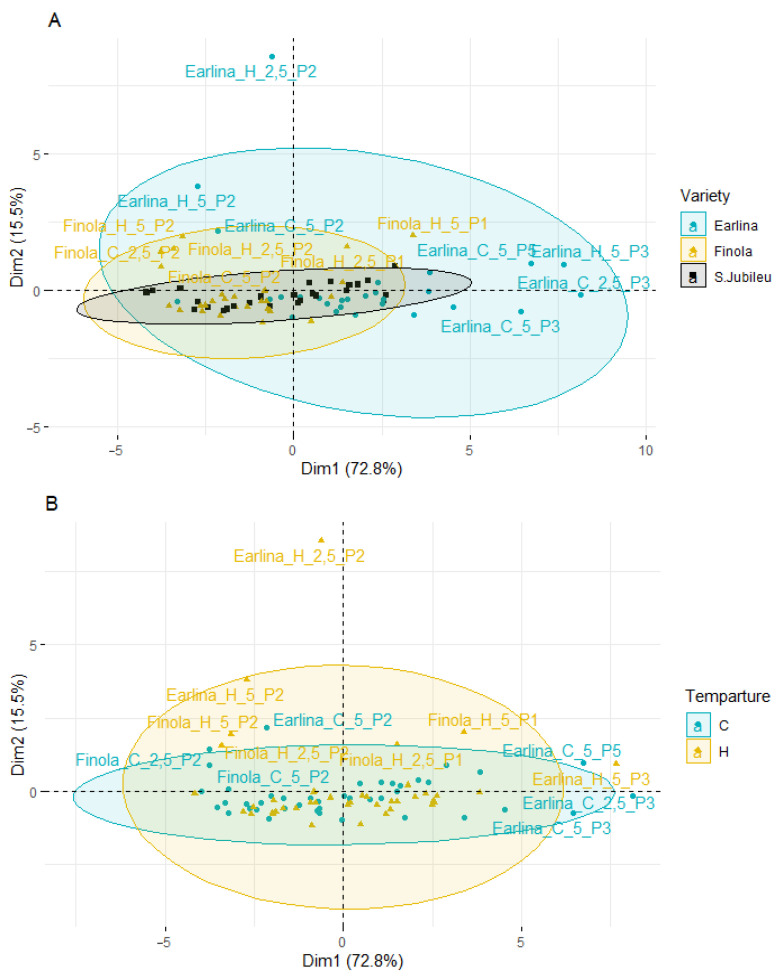
Principal component analysis (PCA) of the score plot of data from phytosterol content in extracted BE (mg/g) varieties (**A**), temperature of pressing process (**B**), type of bleaching earth (**C**), and earth dose (**D**).

**Table 1 molecules-27-07196-t001:** The results of the MANOVA multivariate analysis of variance test for phytosterols’ analysis (Wilks test; *p* < 0.05).

Effect	Value	F	df Effect	df Error	*p*
Free parameter	0.000019	427,585.963	9	75.0000	0.00
Variety (1)	0.000001	7769.1	18	150.0000	0.00
Temperature (2)	0.001233	6752.6	9	75.0000	0.00
Dose of bleaching earth (3)	0.002809	2958.5	9	75.0000	0.00
Bleaching earth (4)	0.000000	1687.4	54	387.0206	0.00
Variety—Temperature	0.000021	1828.8	18	150.0000	0.00
Variety—Dose of bleaching earth	0.010590	72.6	18	150.0000	0.00
Variety—Bleaching earth	0.000000	350.8	108	558.5292	0.00
Temperature—Dose of bleaching earth	0.080871	94.7	9	75.0000	0.00
Temperature—Bleaching earth	0.000000	370.3	54	387.0206	0.00
1–2–3	0.003702	128.6	18	150.0000	0.00
1–2–4	0.000000	418.2	108	558.5292	0.00
1–3–4	0.000000	76.6	108	558.5292	0.00
2–3–4	0.000002	91.6	54	387.0206	0.00
1–2–3–4	0.000000	134.5	108	558.5292	0.00

**Table 2 molecules-27-07196-t002:** Characteristics of bleaching earth.

	**P1**	**P2**	**P3**	**P4**	**P5**	**P6**	**P7**
Foto	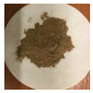	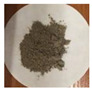	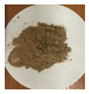	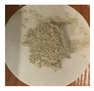	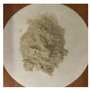	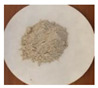	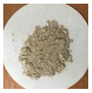
Mineral	Attapulgite clay	Attapulgite clay	Attapulgite clay	Magnesian bentonite	Magnesian bentonite	Magnesian bentonite	Kerolite-hydrated magnesium silicate
Modified	Physically activated	pH-modified	Not modified	Not modified	Not modified	Acid-modified	Chemically modified
pH	8.0	3.2	8.0	8.5	10.0	6.5	6.0
Density (kg/m^3^)	0.41	0.51	0.47	0.60	0.60	0.60	0.55
**Composition (%)**
SiO_2_	55–60	58.8	56.1	63.1	53.5
Al_2_O_3_	2.5	5.3	5.7	8.3	4.0
Fe_2_O_3_	12–14	1.4	1.6	1.9	1.5
MgO	18–21	23.0	23.6	23.0	30.5
CaO	0.5–1.0	2.1	2.7	2.6	0.7
Na_2_O	0.05–0.25	-	-	-	0.3

**Table 3 molecules-27-07196-t003:** Phytosterols content (mg/g) in oils obtained from different hemp varieties and pressing temperature [35].

Phytotserol (mg/g)	Hemp Seeds Oils					
	‘Finola’ (C)	‘Finola’ (H)	‘Earlina 8 FC’ (C)	‘Earlina 8 FC’ (H)	‘S. Jubileu’ (C)	‘S. Jubileu’ (H)
campesterol	0.33 ± 0.00	0.33 ± 0.01	0.32 ± 0.01	0.33 ± 0.02	0.29 ± 0.02	0.29 ± 0.02
campestanol	0.03 ± 0.00	0.03 ± 0.00	0.02 ± 0.00	0.03 ± 0.00	0.03 ± 0.00	0.03 ± 0.00
stigmasterol	0.05 ± 0.01	0.05 ± 0.01	0.04 ± 0.00	0.04 ± 0.01	0.04 ± 0.00	0.03 ± 0.00
β-sitosterol	1.35 ± 0.02	1.28 ± 0.00	1.23 ± 0.09	1.27 ± 0.09	1.26 ± 0.11	1.25 ± 0.07
sitostanol	0.02 ± 0.00	0.04 ± 0.00	0.02 ± 0.00	0.03 ± 0.00	0.02 ± 0.01	0.02 ± 0.01
Δ5-avenasterol	0.16 ± 0.00	0.15 ± 0.02	0.15 ± 0.01	0.16 ± 0.02	0.14 ± 0.02	0.15 ± 0.02
Δ5,24-stigmastadienol	0.06 ± 0.00	0.06 ± 0.01	0.05 ± 0.00	0.06 ± 0.01	0.04 ± 0.01	0.05 ± 0.00
Δ7-avenasterol	0.10 ± 0.00	0.08 ± 0.00	0.08 ± 0.00	0.09 ± 0.01	0.10 ± 0.01	0.05 ± 0.00
24-methylenecycloartenol	0.03 ± 0.00	0.03 ± 0.00	0.03 ± 0.01	0.04 ± 0.00	0.04 ± 0.00	0.04 ± 0.01
TOTAL	2.13 ± 0.03	2.06 ± 0.03	1.95 ± 0.10	2.05 ± 0.15	1.97 ± 0.20	1.97 ± 0.11

C—cold pressing; H—hot pressing.

**Table 4 molecules-27-07196-t004:** The relative proportion of the amount of absorbed compounds from the oil.

BE	TO	P1	P2	P3	P4	P5	P6	P7
‘Finola’ (%)								
2.5	C	0.90	0.69	1.07	0.74	0.56	0.62	0.94
	H	0.95	0.72	0.99	0.64	0.72	0.56	0.90
5.0	C	2.16	1.62	2.14	1.51	1.36	1.26	2.40
	H	2.19	1.58	1.95	1.40	1.40	1.19	1.88
‘Earlina 8 FC’ (%)								
2.5	C	1.01	0.69	1.06	0.61	0.82	0.53	0.76
	H	0.88	0.65	0.85	0.54	0.66	0.52	0.89
5.0	C	2.20	1.49	2.04	1.29	1.66	1.24	1.73
	H	2.41	1.62	2.10	1.31	1.38	1.29	1.96
‘S. Jubileu’ [%]								
2.5	C	0.85	0.62	0.96	0.61	0.69	0.63	0.94
	H	0.96	0.72	0.87	0.57	0.68	0.55	0.89
5.0	C	2.21	1.46	1.94	1.39	1.56	1.40	1.72
	H	2.26	1.04	2.10	1.63	1.46	1.39	1.93

BE—relative share of bleaching earth (%); TO—press method; H—hot; C—cold.

**Table 5 molecules-27-07196-t005:** Weight of extract after methanol extraction.

BE	TO	P1	P2	P3	P4	P5	P6	P7
‘Finola’ (10^−2^ g)								
2.5	C	5.08	5.85	7.10	4.63	5.29	4.84	4.60
	H	4.31	7.54	6.25	4.69	6.14	4.43	4.31
5.0	C	5.53	7.97	7.71	5.18	4.94	5.23	6.18
	H	4.70	9.03	6.74	5.66	5.38	4.36	5.04
‘Earlina 8 FC’ (10^−2^ g)								
2.5	C	5.24	6.54	3.96	3.96	3.78	3.25	3.68
	H	4.40	5.94	3.82	3.49	3.73	3.51	3.99
5.0	C	5.07	6.27	4.17	3.96	3.84	3.91	4.30
	H	4.93	6.95	4.32	3.87	4.02	3.70	4.60
‘S. Jubileu’ (10^−2^ g)								
2.5	C	5.24	6.59	5.66	4.95	6.46	4.74	4.35
	H	6.38	8.80	6.51	6.22	6.09	5.58	5.53
5.0	C	6.56	7.98	6.95	6.15	6.53	5.04	5.41
	H	7.27	7.90	6.80	6.98	6.88	4.66	5.55

BE—relative share of bleaching earth (%); TO—press method; H—hot; C—cold.

## Data Availability

The data presented in this study are available on request from the corresponding author.

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
