# Peer review of "Analysis of Changes in the Amount of Phytosterols after the Bleaching Process of Hemp Oils"

_molecules, 2022, doi:10.3390/molecules27217196_

Round 1

Reviewer 1 Report

Authors of this paper analysed changes in phytosterol contents after the bleaching process of hemp oils. The study appears relevant and could be useful to the affected industries.

My major concerns regarding the quality of the paper include:

(1)    I am not convinced that the authors provided a sound background as to why they singled out phytosterols for analysis. Other important quality and nutritional parameters mentioned in the manuscript need to be determined.

(2)   The manuscript requires significant improvement in the use of English grammar. Minor comments

The use of “numbers of PS” throughout the manuscript is incorrect. Authors could use “individual PS contents” or any other scientific terms.

Line 136: MANOVA is Multivariate analysis of variance, not Multivariate variance

Fig.2. The vertical axis levels representing the Extract weight and Relative size of oil loss should be 0. Not 0,

Fig. 3. The Y axis for PS content should be labeled

Author Response

Dear Reviewer

The research done by our team is generally concerned with the effect of the bleaching process on changing oil parameters. There are unrefined and refined oils available on the market. Niche oils that are available on the market as unrefined. Therefore, there is a need to obtain information on how the bleaching process affects changes in the profile of fatty acids and phytosterols as well as chlorophyll and carotenoids. The bleaching process is a physical process, so it can be recommended for, for example, clarifying oil, as in the case of juices or wine. The results presented here are part of a study, the results of which will be published in future publications due to a large amount of data.

(1)    I am not convinced that the authors provided a sound background as to why they singled out phytosterols for analysis. Other important qualities and nutritional parameters mentioned in the manuscript need to be determined.

I agree with the comment. This publication is exclusively part of a large experiment in which the effect of the bleaching process on Change in color, carotenoid and chlorophyll content, elemental composition and fatty acid profile. Putting all the data at such a level of detail in one article will result in its low clarity. The experiment was divided into five stages. The first publication (characteristics of seeds and hemp oil from these varieties) was published [35]. The next publication will be on the fatty acid profile and composition of the primordia, and the next will be on the color change and carotenoid and chlorophyll changes. The last publication will deal with the realities between all parameters. The results of the experiment include 200 pages of data tables.
This information has been included in the publication.

(2)   The manuscript requires significant improvement in the use of English grammar. Minor comments

Fig.2. The vertical axis levels representing the Extract weight and Relative size of oil loss should be 0. Not 0,

Thank you for pointing out the errors. The publication was fragmented by all co-authors, and the language is at different grammatical levels. We have made corrections in the text as far as possible.

Fig. 3. The Y axis for PS content should be labeled

In response to other comments, the chart has been completely rebuilt, taking into account your comments as well

Reviewer 2 Report

MANUSCRIPT: 1966093

TITLE: Analysis of changes in the amount of phytosterols after the bleaching process of hemp oils

The manuscript 1966093 “Analysis of changes in the amount of phytosterols after the bleaching process of hemp oils”, presents an interesting study in order to find changes in the amount of phytosterols after the bleaching process of hemp oils.

The work is well structured, well planned and the research is competently carried out, the methodology was quite adequate to the research. The literature cited is adequate and most of the papers cited are from the last five years.

The results and discussion are properly discussed although some results are not fully discussed especially due to an incomplete statistical analysis. Conclusions are presented according to the results obtained.

However, some questions remain to be clarified and solved and the manuscript in the current form must be revised in several points as follows comments:

 Results

1.      Please section 2. Results should be called “Results and Discussion”.

2.     Section 2. Results – Please, in order to make this section more attractive for reading, it should be subdivided into subsections according to the different parameters analysed that are described in the material and methods section.

3.     Please put the results referred to in the written sentence between lines 98-99 (Raw vegetable oils, depending on the raw material, are characterized by their individual intense odor and taste) because the odor and taste results are not in the manuscript.

4.     Figure 2 - Please put in the caption of this figure the mention results are the mean ± standard deviation of n =? (indicating the number of replicates of each experience in each parameter determined.

5. Figure 3 – please modify the figure placing the error bars in each parameter and put in the caption of this figure the mention results are the mean ± standard deviation of n =? (indicating the number of replicates of each experience in each parameter determined.

6.     Lines 464, 468 and 470 - Please change “Tabela” by “Table”.

7.     Tables A1, A2 and A3 – Please carry out a comparative statistical analysis to assess for each determined parameter (same column) if are or not there are significant statistical differences due to the different Bleaching earth methods depending on the doses and temperatures used.

8.     Tables A1, A2 and A3 – Please indicate the n (number of replicates of each determination of the different parameters. 

Material and Methods

9.  Please organize the description of materials and methods separately for each of them into subsections and describe each one in detail in order to be able to be reproduced by further studies.

10. Results are presented in the materials and methods section – Please do not present results tables in this section.

11.Table 3 - This table should be removed since they are results that are already published in reference 35 or 36 as mentioned in the text lines 264 and 266. Alternatively, it can be placed in supplementary material.

Other points

12.     Lines 32 and 325 – Please change 2.69%±0.69 by 2.69±0.69 %.

13. Line 30 – Please change 2% by 2 %. Please carefully review all the text and always   give a space between the numerical value and the symbol of the unit of percentage (%).

14. Line 166 - Please change 100ËšC to 100 ËšC. Please carefully review all the text and always give a space between the numerical value and the symbol of the unit of centigrade degree (ËšC).

Author Response

Dear Reviewer

Thank you for pointing out errors in the publication. We have carefully reviewed and made significant changes to the text.

1. Please section 2. Results should be called “Results and Discussion”.

Thank you for the important dodge; in fact, no correction was made to the MDPI formula.

2. Section 2. Results – Please, in order to make this section more attractive for reading, it should be subdivided into subsections according to the different parameters analysed that are described in the material and methods section.

The Results and Discussion section can be divided into two sections: the mass balance of the bleaching process and the results of the sterol profile analysis. A revision has been made in the text.

3. Please put the results referred to in the written sentence between lines 98-99 (Raw vegetable oils, depending on the raw material, are characterized by their individual intense odor and taste) because the odor and taste results are not in the manuscript.

This information is general knowledge. Raw vegetable oils have a characteristic taste and smell, which is removed after refining. Tests of organoleptic parameters were not performed. Only we have a subjective evaluation of taste and odor not resulting from the instrumental analysis.

4. Figure 2 - Please put in the caption of this figure the mention results are the mean ± standard deviation of n =? (indicating the number of replicates of each experience in each parameter determined.

Mean and standard deviation was calculated at n=6. Information is included in the description of Figure 2

5. Figure 3 – please modify the figure placing the error bars in each parameter and put in the caption of this figure the mention results are the mean ± standard deviation of n =? (indicating the number of replicates of each experience in each parameter determined.

Figure 3 has been rebuilt according to your suggestions.

6. Lines 464, 468 and 470 - Please change “Tabela” by “Table”.

Corrected

7. Tables A1, A2 and A3 – Please carry out a comparative statistical analysis to assess for each determined parameter (same column) if are or not there are significant statistical differences due to the different Bleaching earth methods depending on the doses and temperatures used.

Table A1, A2 and A3 are appendices to the publication. Statistical analysis to assess each determinant was presented in the publication. Based on these tables, Figure 2 and Table 1 were created to present the results of the Wilks test.

8. Tables A1, A2 and A3 – Please indicate the n (number of replicates of each determination of the different parameters. 

Each sample measurement was performed three times. Information is completed in the table description.

Material and Methods

  1. Please organize the description of materials and methods separately for each of them into subsections and describe each one in detail in order to be able to be reproduced by further studies

As per your suggestion, we have divided Methods and Materials into subsections. In our opinion, we have described the experiment in a high level of detail. Based on the information in the text, you can reproduce the study. If there is any passage that raises doubts, please specify your comment.

10. Results are presented in the materials and methods section – Please do not present results tables in this section.

The results presented in the Materials and Methods section are the data on which the discussion of the results is based in the Results and Discussion section. Table 3 contains data characterizing the phytosterol profile of the hemp oil used in the bleaching process and is the essence of the information in this chapter. Tables 4 and 5 of the mass balance are the data on the basis of which a descriptive statistical analysis was performed. We insist that these tables remain in this chapter for greater clarity of the presented research results.

11.Table 3 - This table should be removed since they are results that are already published in reference 35 or 36 as mentioned in the text lines 264 and 266. Alternatively, it can be placed in supplementary material.

We wondered about the inclusion of this table, which was already published in the previous publication of the series. We decided that for better clarity of presentation of the results, it should be included in this publication. The publication deals with the change in the phytosterol profile of hemp oils due to bleaching, so it is necessary to present the phytosterol profile of the material used for the experiment.

Other points

  1. Lines 32 and 325 – Please change 2.69%±0.69 by 2.69±0.69 %.
  2. Line 30 – Please change 2% by 2 %. Please carefully review all the textandalways   give a space between the numerical value and the symbol of the unit of percentage (%).
  3. Line 166 - Please change 100ËšC to 100 ËšC. Please carefully review all the text and always give a space between the numerical value and the symbol of the unit of centigrade degree (ËšC).

Thank you for pointing out these spelling errors. A correction has been made in the text.

Reviewer 3 Report

The work is very organized and interesting. However, Minor revisions should be stated.

1-     The following articles should be cited

https://www.mdpi.com/1420-3049/24/1/83

https://aocs.onlinelibrary.wiley.com/doi/full/10.1007/s11745-013-3813-3

2-     In materials,

Time and place for collection of three varieties of hemp seed should be stated.

3-     Add future research plan at the end of the introduction

4-     The authors should explain why they use those three verities only.

5-     Conclusion should not contain reference number. [35] could be replaced by the work of author name and etal.,

Best wishes

Author Response

Dear Reviewer

Thank you for pointing out errors in the publication. We have carefully reviewed and made significant changes to the text. Thank you also for directing us to interesting publications that have broadened the horizon of our knowledge.
Reference to specific comments: 

The following articles should be cited

https://www.mdpi.com/1420-3049/24/1/83

https://aocs.onlinelibrary.wiley.com/doi/full/10.1007/s11745-013-3813-3

Thank you for pointing out interesting literature. Information about them is included in the content of the article.

2-     In materials,

Time and place for collection of three varieties of hemp seed should be stated.

Thank you for pointing out the lack of relevant information. The seed harvesting was done in August 2021 in Poland. The information was posted in the article. 

3-     Add future research plan at the end of the introduction

The results presented here are part of a large project in which the effects of the bleaching process on fatty acid profile, elemental composition, and the degree of color change and reduction of carotenoids and chlorophyll composition were analyzed. Other results will be published in future articles.

4-     The authors should explain why they use those three verities only.

This information was included in the article under the title [35] “Quality of Oil Pressed from Hemp Seed Varieties: ‘Earlina 8FC’, ‘Secuieni Jubileu’ and ‘Finola.’” which was referenced in the text.

5-     Conclusion should not contain reference number. [35] could be replaced by the work of author name and etal.,

I have made modifications to the text, but the reference to the literature list remains for precise indication of publications.

Round 2

Reviewer 1 Report

Authors of the work titled "Analysis of changes in the amount of phytosterols after the 2 bleaching process of hemp oils" have satisfactorily addressed my concerns and have made improvements over the previous version of the manuscript. Hence, I recommend the paper for publication.